# Nanoparticulate air pollution disrupts proteostasis in *Caenorhabditis elegans*

**Bailey A. Garcia Manriquez, Julia A. Papapanagiotou, Claire A. Strysick, Emily H. Green, Elise A. Kikis** [ID] *

Biology Department, The University of the South, Sewanee, TN, United States of America

* eakikis@sewanee.edu

This is a Registered Report and may have an associated publication; please check the article page on the journal site for any related articles.

## Abstract

The proteostasis network comprises the biochemical pathways that together maintain and regulate proper protein synthesis, transport, folding, and degradation. Many progressive neurodegenerative diseases, such as Huntington's disease (HD) and Alzheimer's disease (AD), are characterized by an age-dependent failure of the proteostasis network to sustain the health of the proteome, resulting in protein misfolding, aggregation, and, often, neurotoxicity. Although important advances have been made in recent years to identify genetic risk factors for neurodegenerative diseases, we still know relatively little about environmental risk factors such as air pollution. Exposure to nano-sized particulate air pollution, referred to herein as nanoparticulate matter (nPM), has been shown to trigger the accumulation of misfolded and oligomerized amyloid beta (Aβ) in mice. Likewise, air pollution is known to exacerbate symptoms of AD in people. We asked whether nPM contributes to the misfolded protein load, thereby overwhelming the proteostasis network and triggering proteostasis decline. To address this, we utilized *C. elegans* that express reporter proteins that are sensitive to changes in the protein folding environment and respond by misfolding and displaying readily scorable phenotypes, such as localized YFP fluorescence or paralysis. We found that nPM exacerbated protein aggregation in body wall muscle cells, increasing the number of large visible protein aggregates, the amount of high molecular weight protein species, and proteotoxicity. Taken together, the data point to nPM negatively impacting proteostasis. Therefore, it seems plausible that nPM exposure may exacerbate symptoms of AD and age-related dementia in a manner that is at least partially dependent on proteostasis decline.

## Introduction

The term 'proteostasis' refers to the ability of cells and organisms to maintain a healthy proteome via the activity of the many pathways and processes that comprise the proteostasis network (PN). The proteostasis network consists of nearly 2000 proteins [1] and coordinates the regulation of protein synthesis, folding, trafficking, and turnover [2]. In aging, disease, and under conditions of proteotoxic stress, the PN becomes overwhelmed, resulting in proteostasis

**Data Availability Statement:** All relevant data are within the manuscript and its Supporting Information files.

**Funding:** E.A.K was supported by a Faculty Fellowship from the Appalachian College Association and by two internal grants, a Faculty Development Grant and a Faculty Summer Research Stipend, from the University of the South. B.A.G.M, J.A.P., and C.A.S. were funded by Sewanee Undergraduate Research Fellowships. The CGC, which provides some strains used in this study, and the Emory University Integrated Cellular Imaging Core, which provided microscopy services, were funded by the National Institutes of Health. Funders had no role in study design, data collection and analysis, decision to publish, or preparation of the manuscript.

**Competing interests:** The authors have declared that no competing interests exist.

collapse and an increased load of misfolded proteins [2]. This proteostasis collapse has been documented both in young *C. elegans* engineered to express misfolded Huntington's disease (HD)-associated protein [3] and during aging [4].

HD is a progressive autosomal dominant neurodegenerative disorder for which the genetic determinant is an expansion of a polyglutamine (polyQ)-encoding CAG repeat in the gene that encodes the huntingtin protein [5]. PolyQ-expanded huntingtin protein misfolds and aggregates in aging individuals, and is neurotoxic.

As in HD pathogenesis, protein aggregation is also a hallmark of Alzheimer's disease (AD). For example, the amyloid precursor protein (APP) is misprocessed into a misfolded and aggregation-prone neurotoxic peptide, referred to herein as amyloid beta (Aβ). The misfolding/aggregation of Aβ precedes tau hyperphosphorylation, the formation of neurofibrillary tangles, and the onset of cognitive decline [6]. Genome-wide association studies have uncovered 29 genetic risk factors for sporadic AD [7]. ApoE4, identified in 1993, is the strongest such risk factor, associated with a two- to ten-fold increased risk for the disease compared to the general population. It likely acts via an interaction with Aβ [8, 9], whose misfolding and deposition within specific regions of the brain may partially contribute to neurodegeneration [10].

Environmental risk factors for AD are also under intense investigation, pointing to the need to consider the role that gene—environment interactions play in disease progression. Underscoring the disease-relevance of gene—environment interactions, a recent epidemiological study revealed that particulate air pollution significantly exacerbates the effects of ApoE4 in women [11]. Likewise, in a mouse model of familial AD, the presence of the human ApoE4 allele was associated with increased susceptibility to nano-sized particulate matter (nPM) obtained from traffic-derived air pollution, leading to an increase in Aβ aggregation [11]. These important findings suggest that the ability to maintain proteostasis is likely compromised upon exposure to particulate air pollution.

Consistent with this hypothesis, it was recently shown that the expression levels of proteostasis network genes in *C. elegans* are responsive to nPM [12]. Furthermore, the degradative pathways of the proteostasis network have also been shown to be activated in mice exposed to nPM [13]. The effects of this dysregulation on the folding, or misfolding, of disease-associated proteins has never been directly tested. Here we address the hypothesis that exposure to nPM may challenge the buffering capacity of the proteostasis network, thereby reducing the efficiency of disease-associated protein folding.

It is important to note that neuroinflammation is induced upon nPM exposure [14], making it difficult to ascertain whether effects of nPM on proteostasis are upstream or downstream of inflammation. *C. elegans* lack the transcription factor NFκB and thus do not experience a canonical inflammatory response. Therefore, by utilizing *C. elegans* as our model system, we were able to separate these two processes and examine the impact of particulate air pollution on proteostasis without the confounding effects of inflammation. Furthermore, the tools and sensors required to monitor changes in *C. elegans* proteostasis in real-time are plentiful and well-documented, making this animal an ideal model for our purposes. Specifically, we utilized wild type *C. elegans* and strains that have been engineered to express aggregation-prone disease-associated proteins, including polyQ and Aβ peptides, as sensors of the protein folding environment.

We found that exposure to nPM exacerbated the misfolding and toxicity of polyQ35::YFP and Aβ in body wall muscle cells, implying that nPM contributes to an increase in the misfolded protein load. Assuming that this effect translates to humans, the increased protein misfolding suggests that nPM exposure may ultimately increase disease risk by causing the protein quality control machinery to become overwhelmed by damaged protein, thus disrupting the delicate proteostasis balance.

**Table 1.**

| Strain Name | Genotype | Reference |
|---|---|---|
| N2 | Wild isolate (Bristol) | |
| AM140 | unc-54p::Q35::YFP:: unc-54 3'-UTR | (Morley, Brignull et al. 2002) [16] |
| AM141 | unc-54p::Q40::YFP:: unc-54 3'-UTR | (Morley, Brignull et al. 2002) [16] |
| GMC101 | unc-54p::A-beta-1-42::unc-54 3'-UTR | (McColl, James et al. 2012) [17] |
| OG412 | vha-6p::Q44::YFP + rol-6(su1006) | (Mohri-Shiomi, et al. 2008) [18] |

## Materials and methods

### *C. elegans* strains and culture

The strains utilized in this study (**Table 1**) were obtained from the *Caenorhabditis elegans* Genetics Center (CGC) at the University of Minnesota and maintained at 20°C on Nematode Growth Media (NGM) seeded with OP50 bacteria as a food source according to the standard methods [15]. Age-matched animals were synchronized within a 4hr developmental window obtained via egg lays.

### Acquiring nanoparticulate matter from polluted air

Traffic-derived nanoparticulate air pollution samples (nPM) were collected in Los Angeles, California using a high-volume ultrafine particulate (HVUP) sampler with a Teflon filter. Dried nPM samples were eluted to 150 μg/mL with deionized water according to established methods [19]. Characterized and validated nPM samples have been generously gifted to us by the laboratory of Caleb Finch of the University of Southern California. As bacterial contamination could affect proteostasis, nPM samples were sterilized by UV-C irradiation in a biosafety cabinet for at least 15min. To ensure that samples were free of bacterial contamination, 20μL of nPM was transferred to an NGM plate, incubated for 3 days, and examined for the appearance of colonies. For consistency, a single batch of eluted nPM was used for all of the experiments proposed herein.

### Exposure paradigm

*C. elegans* were grown to the L1 or L4 stage (as specified below for each assay), at which point at least 20 animals were exposed to 75μg/mL nPM. The exact number of animals exposed depended on the constraints of specific experiments and is specified for each assay. M9 served as our negative control, while 5mM paraquat (PQ) was used as a positive control for oxidative stress. Exposures were performed at 20°C in 96 well plates. 100μL of 2X M9 liquid medium was diluted with an equal volume of nPM (paraquat, and/or water) supplemented with 10μg/mL cholesterol and OP50 bacteria and incubated at 20°C for up to 3d. To maintain uniform nPM concentration for the duration of the experiment, and to prevent animals from falling to the bottom of the well, 96-well plates underwent continuous gentle rocking on a nutator. All exposures were performed at least in biological triplicate. Bagged animals or those that were otherwise visibly unhealthy were eliminated from further analyses.

### Counting aggregates

**Live imaging of YFP-tagged proteins.** For YFP-tagged AM140 and OG412 animals, exposures were initiated at the L4 larval stage. For AM141 animals, exposures were initiated at the L1 larval stage. All animals were chronically exposed for 72h hours. After exposure,

AM140 and AM141 animals were allowed to recover for ~1h on NGM plates prior to imaging. For OG412, animals were allowed to recover for 3d on NGM prior to imaging to ensure sufficient time for the age-dependent formation of large visible aggregates in the intestine. Imaging was performed with a Leica M165FC stereomicroscope (Welzlar, Germany) fitted with an ORCA-Flash 4.0 V3 camera (Hamamatsu Photonics, Japan). Prior to imaging, live animals were chilled on ice to limit movement.

**Immunofluorescence.** GMC101 animals were exposed to M9, nPM, or paraquat for 24h starting at the L4 stage followed by transfer to NGM plates seeded with OP50 and incubation at 25°C for 1d. Detection of Amyloid Beta (Aβ) protein, which is not fluorescently tagged, was performed via immunofluorescence. In short, animals were fixed with 2% paraformaldehyde, treated with β-mercaptoethanol, and permeabilized with collagenase prior to immunostaining as previously described [3]. Aβ was probed with the monoclonal anti-amyloid beta antibody derived from clone BAM-10 sold by Sigma-Aldrich (St. Louis, MO). The secondary antibody was conjugated to an Alexa Fluor® 488 (ab150113) from Abcam (Cambridge, United Kingdom) for visualization on a Leica SP8 laser scanning confocal microscope (Welzlar, Germany) using a 40X oil objective. All aggregation assays were performed in biological triplicate for a total of ~20 animals (n = 20) per genotype per exposure.

## Thrashing assays

For AM140 animals, exposures were initiated at the L4 larval stage, whereas exposures were initiated at the L1 larval stage for AM141 animals. Both strains were chronically exposed for 72h hours and allowed to recover for ~1h on NGM plates prior to measuring the thrashing rate. To assay for changes in proteotoxicity triggered by exposure to nPM, thrashing assays were performed as described previously [20]. In short, single animals were placed on a drop of M9 on a glass slide and, following a 30s recovery period, the number of body bends per minute was counted. Thrashing assays were performed in biological triplicate with a total of ~20 animals (n = 20) per genotype per exposure.

## Paralysis assays

To assay for changes in Aβ toxicity in body wall muscle cells in response to nPM, GMC101 animals were exposed to +/- nPM either as L4s for 1hr (acute stress) or as L1s for 3d (chronic stress). Paralysis was then monitored at 25°C for at least 3 days. N2 animals were utilized as a control to determine whether any observed paralysis is due to a gene (Aβ)—environment (nPM) interaction or is simply an effect of the nPM, irrespective of Aβ. All paralysis assays were performed in at least biological triplicate.

## Native gel electrophoresis

Native protein from 50 animals was extracted mechanically by grinding in liquid nitrogen followed by the addition of 30μL of ice-cold native lysis buffer as described previously (24). Native samples were resolved immediately after extraction on a 6% native PAGE gel. Fluorescent bands containing YFP protein were detected under UV light on a BioRad gel-doc imaging system (Hercules, CA). YFP-containing polyQ protein bands representing either monomeric protein or high molecular weight species were quantified for each lane using ImageJ and the ratio of monomers to high molecular weight species in each lane was calculated and graphed as means of three experiments (biological triplicate).

Detection of unlabeled Aβ oligomers was as described above for polyQ, but with some modifications. Specifically, after native gel electrophoresis, gels were heated in SDS to denature the resolved protein and then a western transfer to a PVDF membrane was performed as

described [21]. Standard immunoblot protocols with anti-Aβ antibodies (clone BAM-10, Sigma) were intended to allow visualization using the LI-COR Odyssey system (Lincoln, NE). Unfortunately, this antibody and an antibody from Cell Signaling (D3E10) failed to provide a reliable signal and thus the Aβ native gels are omitted from this registered report.

### Western blot analysis

Total protein was obtained from 10 animals by boiling whole worms in Laemmli sample buffer for 5 min prior to loading on a 10% sodium dodecyl sulfate-polyacrylamide gel (SDS-PAGE). Following electrophoresis, proteins were transferred to a PVDF filter. Immunodetection was with an RDye800-conjugated anti-GFP antibody (Rockland Immunochemicals, catalog. No. 600-132-215, Gilbertsville, PA). Imaging was with the Li-Cor Odyssey system (Lincoln, NE). As for the native gels, the anti-Aβ antibodies did not yield a signal on western blots and, thus, the planned westerns are omitted. PolyQ::YFP western blots were performed in triplicate and band intensity was quantified with Image J software (version 1.8.0).

### qRT-PCR

RNA was isolated from 20 animals at 1h, 24h, or 72h post-exposure using 250μL Trizol (Sigma-Aldrich, St. Louis, MO) according to the manufacturer's instructions. Removal of genomic DNA and cDNA synthesis was performed with the iScript gDNA clear cDNA synthesis kit (Bio-Rad, Hercules, CA). qPCR was performed with the MyiQ single-color real-time PCR detection system with SYBR green master mix (Bio-Rad, Hercules, CA) using previously published gene-specific primers (Table 2). To control for differences in sample concentration, the expression of stress genes was normalized to actin. Gene expression was further normalized to the 1h M9 control and pairwise T-tests were performed at each time point to compare each exposure to the control. Data represent averages of six biological replicates.

### Graphs and post-hoc analyses

All graphing and statistical analyses were performed via GraphPad Prism 7 (San Diego California). Unless otherwise noted, bars represent means, error bars represent standard error of the mean (SEM), and * indicates a p-value less than 0.05 from pairwise T-tests with Welch's correction. $IT_{50}$ for paralysis assays refers to the time it took for 50% of the GMC101 animals to experience paralysis under different conditions (+/-nPM). These values were determined via a linear regression analysis or non-linear fit of normalized, non-logarithmic, data.

## Results and discussion

### Nanoparticulate matter from traffic-derived air pollution exacerbates protein aggregation

Proteostasis sensors, comprising aggregation-prone disease-associated proteins, have been previously developed for *C. elegans* and used to determine that the proteostasis network (PN)

**Table 2. Primers used to investigate changes in gene expression.**

| Gene | Forward Primer | Reverse Primer | References |
|---|---|---|---|
| hsp-4 | CTAAGATCGAGATCGAGTCACTC | GCTTCAATGTAGCACGGAAC | Haghani et. al., 2019 [12] |
| gst-4 | GATGCTCGTGCTCTTGCTG | CCGAATTGTTCTCCATCGAC | Haghani et. al., 2019 [12] |
| hsp-6 | TCGTGAACGTTTCAGCCAGA | CTCAGCGGCATTCTTTTCGG | Bennet et. al., 2014 [22] |
| C12C8.1 | ACGGGCTTTCCTTGTTTT | ACTCATGTGTCGGTATTTATC | Prahlad et. al., 2008 [23] |
| F44E5.4 | TGTCCTTTCCGGTCTTCCTTTTG | AATGAACCAACTGCTGCTGCTCTT | Prahlad et. al., 2008 [23] |
| Actin | ATCACCGCTCTTGCCCCATC | GGCCGGACTCGTCGTATTCTT | Prahlad et. al., 2008 [23] |

is easily overwhelmed, even by slight changes in the abundance of misfolded protein [3, 4]. We used these established sensors to ask whether traffic-derived air polution causes an increase in protein damage that is sufficient to disrupt proteostasis in *C. elegans*. Specifically, we exposed *C. elegans* that express polyQ peptides or the Aβ peptide in body wall muscle cells or intestines to three different conditions. One condition was nanoparticulate matter, which was derived from vehicular emissions (nPM), one was paraquat (PQ), which served as a control for oxidative stress, and the other was simply the vehicle, M9. Protein aggregation was then monitored based on the amount or number of large visible Aβ or polyQ aggregates.

Specifically, animals expressing either polyQ35::YFP or polyQ40::YFP in body wall muscle cells were chronically exposed to nPM or PQ for 3d in liquid culture or mock exposed to the vehicle M9. After that time, fluorescent aggregates were imaged and counted. The average number of polyQ35 aggregates was not altered in response to PQ but increased 1.6-fold in response to nPM (**Fig 1A and 1B**). In contrast, polyQ40::YFP aggregation decreased slightly in nPM (**S1A Fig**). These apparently contradictory results can likely be explained by exposures being performed at different times in the developmental cycle of the animal or by polyQ35:: YFP being a better sensor than polyQ40::YFP to detect subtle changes in the protein folding environment. We generally favor the latter hypothesis because polyQ40::YFP is already highly aggregated even in young animals, making changes to aggregation subtle and thus difficult to detect. In contrast the polyQ35::YFP, having a slightly shorter polyQ tract length, is closer to the aggregation threshold. As a result, polyQ35::YFP has very few aggregates in young adults, but becomes progressively more aggregated in an age-dependent manner [16] and in a manner that is responsive to changes in molecular chaperone abundance [24], making it a powerful sensor of slight changes in proteostasis. Similar to what we observed in body wall muscle cells, aggregation of polyQ44::YFP in intestinal cells, one of the first internal tissues in contact with nPM, increases slightly after nPM exposure (**S1B Fig**).

Because aggregation is protein concentration-dependent [25], it was important to rule out increased aggregation being due to an increase in protein expression. To that end, we measured steady-state levels of our reporter proteins via Western blot analysis. We observed no statistically significant differences in protein levels (**S2 Fig**), indicating that the changes we observed in aggregation are most likely due to a declining ability to maintain proteostasis under conditions of nPM exposure.

Because nPM is known to exacerbate Alzheimer's disease symptoms in people and Aβ aggregation in mice [11], we asked whether, similar to what we observed with polyQ35, Aβ expressed in *C. elegans* body wall muscle cells also aggregates in response to nPM. The large number of Aβ aggregates made determining the exact number in individual animals impossible. However, confocal microscopy of the heads of individuals exposed to nPM or PQ and stained with an anti-Aβ antibody revealed what appears to be an increase in both tiny Aβ puncta and larger aggregates (**Fig 1C**). Taken together with the aforementioned polyQ35 aggregation enhancement, these data suggest that nPM from traffic-derived air pollution exacerbates protein misfolding in *C. elegans*, especially in body wall muscle cells. This is consistent with nPM triggering a decline in proteostasis, likely by causing sufficient protein damage as to overwhelm the protein quality control machinery.

## Nanoparticulate matter from traffic-derived air pollution is associated with proteotoxicity

Increased aggregation of polyQ in *C. elegans* body wall muscle cells is typically associated with proteotoxicity, resulting in observable motility defects [16]. We thus predicted that the observed increase in protein aggregation that occurred upon exposure to nPM would likewise

**A**

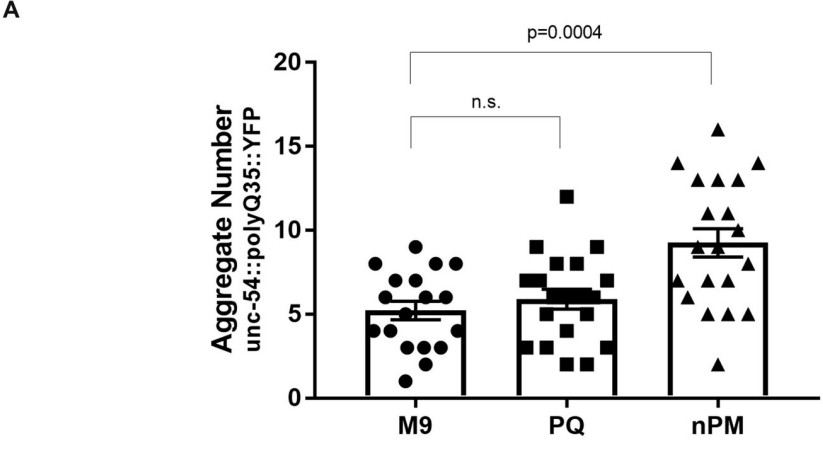

**B**

**C**

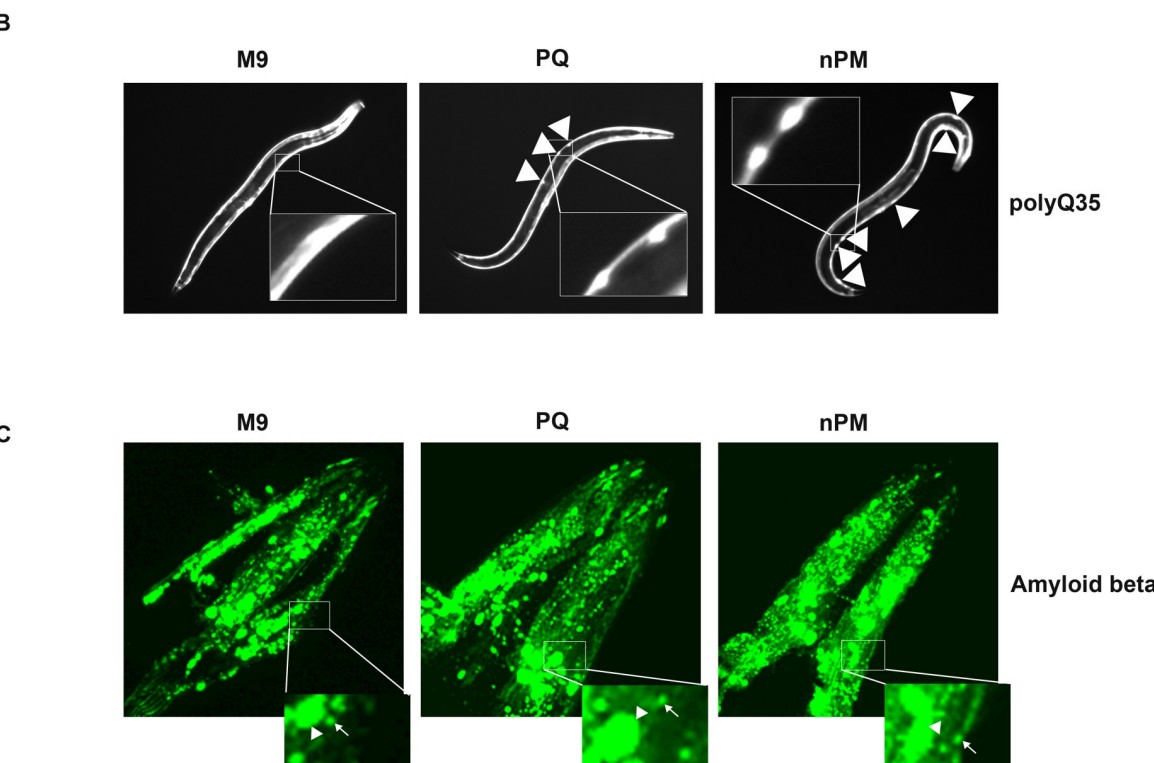

**Fig 1. Exposure to nPM results in increased polyQ aggregation.** (A and B) *C. elegans* expressing polyQ35::YFP in body wall muscle cells (strain AM140) were exposed for 3d to nanoparticulate matter (nPM), the oxidant paraquat (PQ), or mock exposed to vehicle (M9) starting at the L4 stage. A) Graph representing the number of large visible aggregates in body wall muscle cells. The data represent biological triplicates with all individuals indicated (●, M9; ■, PQ; ▲, nPM). Bars represent means and error bars represent the standard error of the mean (SEM). P-values are the results of T-tests with Welch's correction. "n.s." refers to differences that are not statistically significant. B) representative fluorescent micrographs of living polyQ35::YFP animals on NGM plates 3d after the onset of exposures. Arrows point to visible aggregates. Inset shows magnification of region marked with a rectangle in order to more clearly depict aggregates. C) Confocal micrographs of the head regions of animals expressing amyloid beta (Aβ) in body wall muscle cells (strain GM101). L4 stage animals were exposed for 24h to nPM or PQ and then transferred to 25°C where they remained until day 3 of adulthood, prior to fixation and staining with an anti-Aβ antibody (BAM-10, Sigma-Aldrich, St. Louis, MO). Inset shows magnification of region marked with a rectangle. Within the inset, a representative large aggregate (arrowhead) and a representative small punctum (arrow) is indicated for each exposure. Exposures were performed in biological triplicate, with ~20 animals examined per exposure per triplicate.

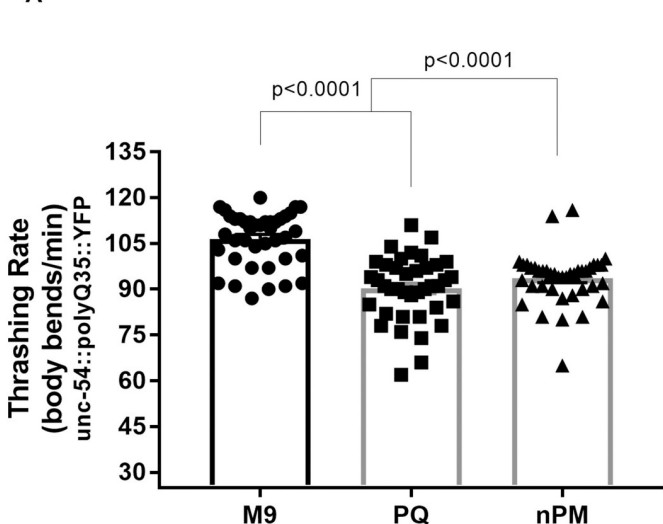

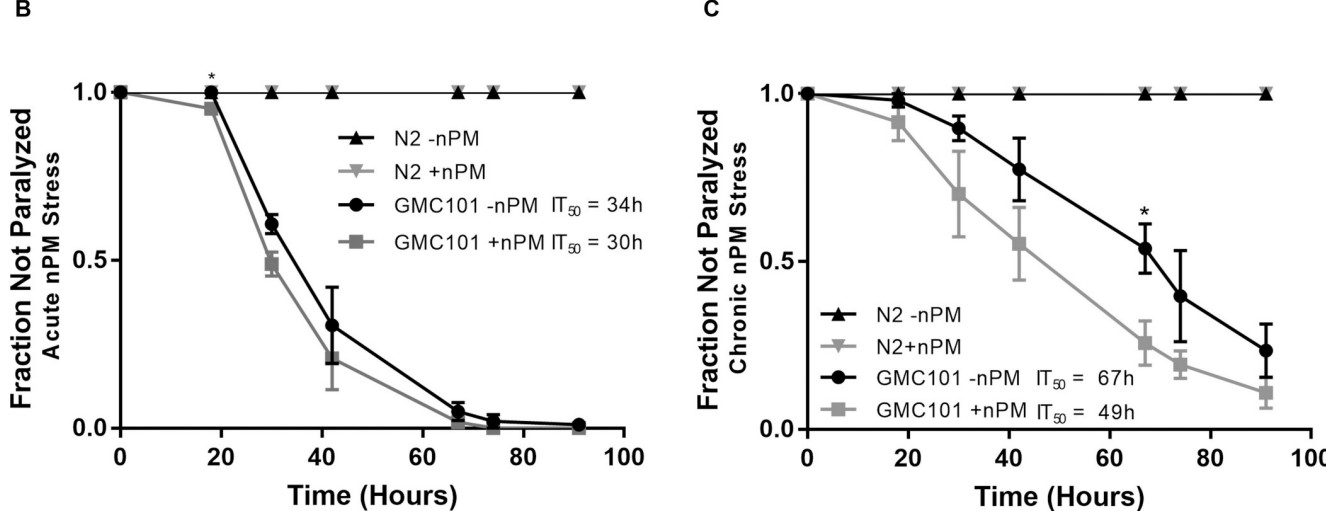

**Fig 2. nPM is associated with increased toxicity.** A) Toxicity was measured as a function of thrashing rate in liquid. *C. elegans* expressing polyQ35::YFP in body wall muscle cells were exposed for 3d to nanoparticulate matter (nPM), the oxidant paraquat (PQ), or mock exposed to vehicle (M9) starting at the L4 stage. Bars represent the average number of body bends per minute. The experiment was performed in biological triplicate with all individual measurements indicated (●, M9; ■, PQ; ▲, nPM). Error bars represent the standard error of the mean (SEM) and P-values are the results of pairwise T-tests with Welch's correction. B and C) L4 stage adult wild type animals (N2) or those expressing amyloid beta (Aβ) in body wall muscle cells (strain GM101) were exposed to nPM, or M9 as a control, for 1h (B) or 72h (C) and then transferred to NGM plates seeded with OP50. Paralysis was monitored from that point forward for 96hrs at the restrictive temperature of 25°C. Data represent averages of biological triplicates, error bars represent the standard error of the mean. * represents $p < 0.05$ for pairwise T-tests, which were preformed to determine statistically significant differences in paralysis between +/-nPM at each timepoint individually. $IT_{50}$ refers to time to 50% paralysis for GMC101 animals exposed to +/- nPM as determined following regression analysis. Differences between $IT_{50}$ for +/-chronic nPM exposure (C) are statistically significant ($p = 0.0003$).

result in decreased motor function. Consistent with this expectation, exposure of polyQ35::YFP animals to the oxidant PQ resulted in a 16% decrease in thrashing rate while exposure to nPM decreased the thrashing rate by 13% (**Fig 2A**). A similar trend was also observed for polyQ40::YFP animals, which experienced a 7% decline in thrashing rate in response to nPM (**S3 Fig**) even though they did not show increased aggregation (**S1A Fig**). The finding that

toxicity was observed in polyQ40::YFP animals in response to nPM without a concomitant increase in protein aggregation may be explained by nPM exerting a toxic effect on muscle cells in a manner that is independent of protein folding/aggregation, or by the effect of PQ or nPM on protein folding being outside our level of detection, especially when using the less sensitive polyQ40::YFP.

Aβ expressed in body wall muscle cells has been shown to cause a temperature-dependent decline in motor function, resulting in paralysis over time [26]. To determine whether this effect is exacerbated by nPM, we exposed Aβ animals (GMC101) to nPM for either 1h (acute exposure) or 3d (chronic exposure) and then moved them to the restrictive temperature of 25˚C. Paralysis at 25˚C was monitored at regular intervals over the course of 3d. We found that both acute and chronic exposure to nPM increased the toxic effects of Aβ, with statistically significant decreases in motility at some timepoints according to pairwise T-tests. It is noteworthy that, in contrast to the paralysis observed in Aβ animals, wild type (N2) animals remained mobile for the duration of the analysis regardless of the exposure paradigm. This implies that nPM is not toxic to wild type animals and further suggests that the observed paralysis in GMC101 is dependent on the presence of Aβ.

To determine whether the overall paralysis time courses are statistically significantly different in +/-nPM conditions, we performed regression analysis to compare $IT_{50}$ (time to 50% paralysis) for each condition. The acute nPM stress data fit non-linear curves with $R^2 = 0.7$ for both the +/-nPM data. Under these conditions, we found that paralysis occurred earlier, with an $IT_{50}$ decreasing from 34h to 30h upon acute nPM exposure (**Fig 2B**) although with SE of ~5.5, these differences are not statistically significant. The data obtained under chronic stress conditions best fit a linear regression model ($R^2 < 0.8$ for both data sets) and a replicates test for lack of fit validated the linear model. With $p = 0.0003$, the differences between the chronic +/-nPM lines of best fit are significantly different and reveal that $IT_{50}$ decreased substantioally from 67h to 49h upon chronic nPM exposure (**Fig 2C**), indicative of earlier onset on proteostasis decline.

Together, the data from polyQ35::YFP animals and Aβ animals suggest that exposure to nPM increases the load of misfolded protein, which, in turn, leads to a decline in cellular function. As both of these proteins were expressed in body wall muscle cells, this cellular decline manifests as motility defects.

## nPM is associated with an increase in the relative amount of high molecular weight protein species

The toxic oligomer hypothesis posits that the large visible aggregates such as those described herein (**Fig 1**) may be cytoprotective. Instead, smaller aggregated species such as oligomers, which are on-pathway to becoming large aggregates, may be the toxic species [27]. To determine whether exposure to nPM causes an increase in the relative amount of high molecular weight species (presumptive oligomers), we performed native gel electrophoresis. This allowed us to resolve large aggregates that remain in wells, high molecular weight species that enter the gel, and monomers that run toward the bottom of the gel. We found that exposure to nPM significantly increased the relative abundance of high molecular weight polyQ35::YFP species relative to monomers (**Fig 3**). By contrast, nPM had little to no effect on the observed abundance of high molecular weight polyQ40::YFP species relative to monomers (**S4A and S4B Fig**), likely because its high basal amount of aggregation meant that the baseline presence of monomers was so low as to be at or below the limit of detection. On the other hand, polyQ44::YFP in intestinal cells was more responsive to PQ than to nPM with respect to the relative abundance of high molecular weight species (**S4C and S4D Fig**), suggesting that intestinal protein aggregation dynamics may be especially sensitive to oxidative stress.

A                                                    B

**Fig 3. nPM causes an increase in oligomeric species.** *C. elegans* expressing polyQ35::YFP in body wall muscle cells (strain AM140) were exposed for 3d to nanoparticulate matter (nPM), the oxidant paraquat (PQ), or mock exposed to vehicle (M9) starting at the L4 stage and total protein was extracted into native lysis buffer via mechanical disruption. A) Representative native gel depicting in-gel YFP fluorescence with monomeric and high molecular weight species indicated. B) Quantification of high molecular weight species relative to monomers calculated as the ratio of the intensity of the two indicated bands in each lane. Bars represent means of biological triplicates with individual measurements shown (●, M9; ■, PQ; ▲, nPM). Error bars represent the standard error of the mean. P-values are the results of T-tests performed with Welch's correction. "n.s." refers to differences that are not statistically significant.

## Chronic exposure to nPM does not induce a robust stress response

A prior study revealed that *gst-4* protein levels increased in L1 or L4 stage animals in response to 1h of nPM exposure, although a robust transcriptional response was only observed in L1 animals [12]. Additionally, *hsp-4* gene expression was shown to decrease in L1 animals following exposure to nPM for 1h [12]. To determine whether chronic nPM exposure induces a transcriptional stress response, we exposed L4 animals to nPM or PQ for 3 days and examined stress gene expression at 1h, 24h, and 72h of exposure. Specifically, we examined the expression of *gst-4* as a readout of oxidative stress, *hsp-4* as a readout of ER stress, *hsp-6* as a readout of mitochondrial stress, and the Hsp70s C12C8.1 and F44E5.4 as two separate readouts of the cytosolic heat shock response. We found that while PQ induced the expected oxidative stress response, with statistically significant upregulation of *gst-4* gene expression at both 24h and 72h post exposure (**Fig 4A**), nPM had no measurable effect on any stress genes examined (**Fig 4A–4E**), indicating that the canonical stress responses in the cytosol, ER, or mitochondria were not engaged by nPM. It is possible that the combined challenge of nPM and an aggregation-prone disease-associated protein such as polyQ or Aβ may be sufficient to induce a stress response, whereas nPM alone is clearly insufficient. Alternatively, it may be that without robust cytoprotective stress responses, the buffering capacity of the proteostasis network remains low upon npM exposure, such that disease-associated protein aggregation/toxicity is exacerbated. These hypotheses can be tested in future studies by examining the effects of nPM on transcriptional stress responses in the polyQ35::YFP or polyQ40::YFP genetic backgrounds.

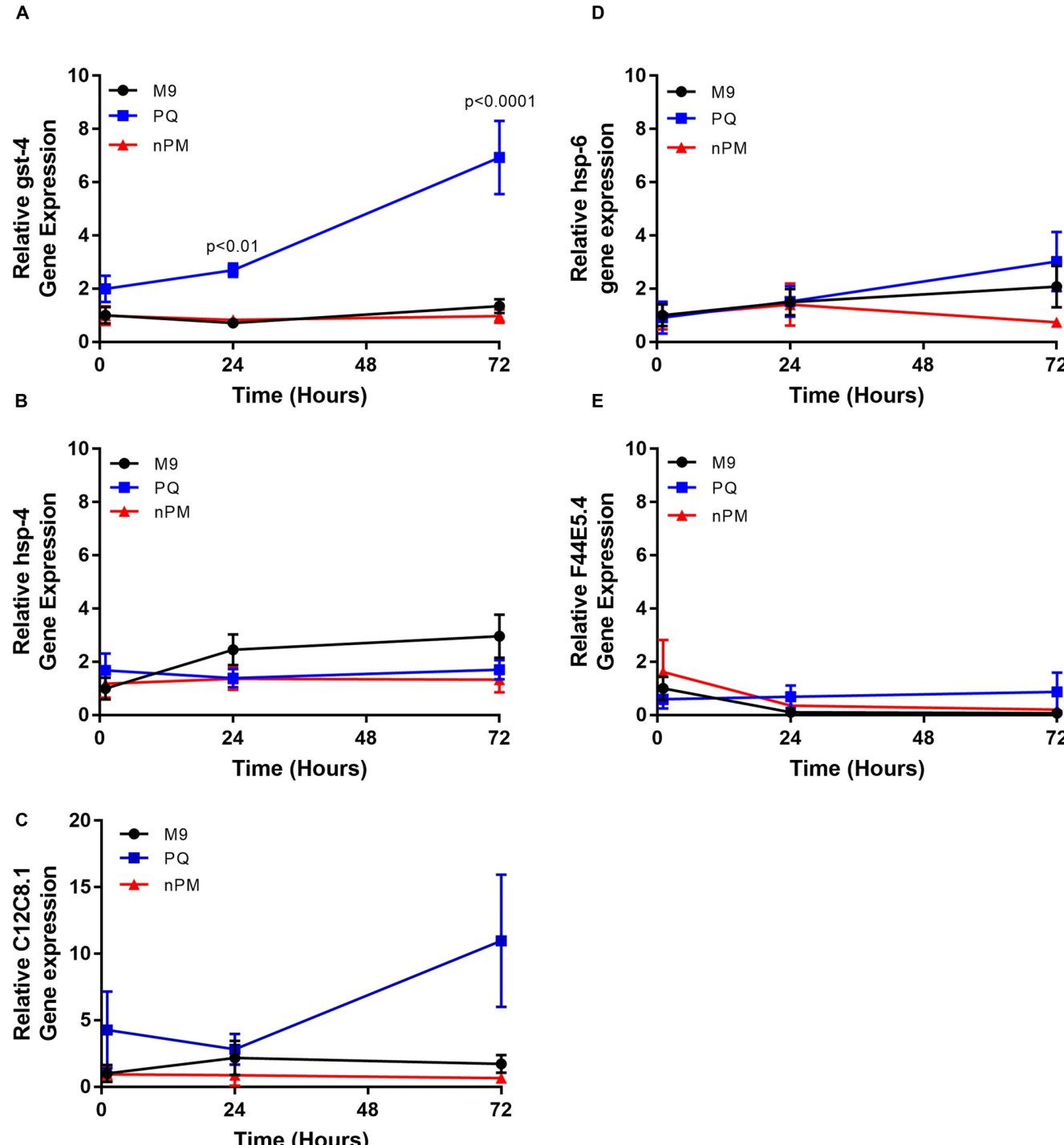

**Fig 4. nPM does not induce a robust transcriptional stress response.** N2 animals were exposed for 3d to nanoparticulate matter (nPM), the oxidant paraquat (PQ), or mock exposed to vehicle (M9) starting at the L4 stage. RNA was isolated from 20 animals at 1h, 24h, or 72h post-exposure and gene expression was determined by qRT-PCR. Data points represent average fold change relative to M9 samples at 1h of exposure and error bars represent the standard error of the mean of six biological replicates. Statistically significant differences relative to the mock (M9) control at each time point were determined by T-tests. P-values are indicated only in cases where $p < 0.05$.

## Conclusions

That air pollution is an environmental risk-factor for Alzheimer's disease is well-established (11) and its mechanism of action is under investigation. Evidence points to either a direct effect of nPM on proteostasis or an indirect effect through triggering neuroinflammation (14). Because *C. elegans* lack an inflammatory response, we used it to ascertain whether nPM is able to cause proteostasis imbalance in a manner independent of inflammation. Our findings point to nPM having a modest negative impact on proteostasis in *C. elegans* consistent with an inflammation-independent mode of action. Specifically, nPM exacerbated the aggregation of polyQ35 and Aβ in body wall muscle cells (**Fig 1**), and decreased the relative amount of mono-meric polyQ35 (**Fig 3**). Likewise, nPM increased the toxicity of both of these established pro-teostasis sensors (**Fig 2**). Because nPM disrupts proteostasis in *C. elegans*, it seems theoretically plausible that nPM exposure may worsen Alzheimer's disease symptoms and age-related dementia, at least in part, by directly overwhelming the proteostasis network and thereby dis-rupting the delicate proteostasis balance. This may be especially true for olfactory neurons, which are in direct contact with the air and which experience pathological protein misfolding early in Alzheimer's' disease progression [28].

## Supporting information

**S1 Fig. PolyQ protein aggregation in response to nPM.** A) *C. elegans* expressing polyQ40::YFP in body wall muscle cells (strain AM141) were exposed for 3d to nanoparticulate matter (nPM), the oxidant paraquat (PQ), or mock exposed to vehicle (M9) starting at the L1 stage. Because L1 stage animals experienced low survival in PQ, they are omitted from this analysis. B) Animals expressing polyQ44::YFP in intestinal cells (strain OG412) were exposed to nPM for 72hrs starting at the L4 stage. After exposures, animals were allowed to recover for 3d on NGM plates seeded with OP50 at which time aggregate number was determined. Graphs depict the number of large visible aggregates in either body wall muscle cells (A) or intestinal cells (B). Bars represent the average number of aggregates with the number of aggregates in each individual also indicated (●, M9; ■, PQ; ▲, nPM). Error bars represent the standard error of the mean (SEM). P-values are the results of T-tests with Welch's correction. "n.s." refers to differences that are not statistically significant. All exposures were performed in bio-logical triplicate.
(TIF)

**S2 Fig. nPM does not alter the steady-state levels of polyQ protein.** All animals were exposed for 3d to nanoparticulate matter (nPM), the oxidant paraquat (PQ), or mock exposed to vehicle (M9). Exposures started at the L4 stage for polyQ35::YFP animals (strain AM140) (A, D), at the L1 stage for polyQ40::YFP animals (strain AM141) (B, E), and at the L4 stage for the polyQ44::YFP animals (strain OG412) (C, F). Immunoblots of total protein were probed with an anti-GFP antibody. (A, B, C) Graphs represent the average amount of YFP-containing protein in biological triplicates. Error bars represent the standard error of the mean (SEM). All possible pairwise T-tests were performed with Welch's correction and no statistically signifi-cant differences in steady-state protein levels were observed between treatments. P-values are only shown for polyQ44::YFP because that strain had the greatest amount of variability in western blot analysis. (D, E, F) Representative immunoblots. The doublet marked with a * likely represents YFP alone as the result of proteolysis between the polyQ40 and YFP moieties during sample preparation. Both upper and lower bands were included in the quantification.
(TIF)

**S3 Fig. nPM is associated with increased polyQ40 toxicity.** *C. elegans* expressing polyQ40::YFP in body wall muscle cells (strain AM141) were exposed for 3d to nanoparticulate matter (nPM), the oxidant paraquat (PQ), or mock exposed to vehicle (M9) starting at the L1 stage. Toxicity is represented as a function of thrashing rate in liquid measured 30min after exposure. Assays were performed in biological triplicate and the graphs represent the average thrashing rate with individual measurements indicated (●, M9; ■, PQ; ▲, nPM). Error bars represent the standard error of the mean (SEM). P-values are the results of T-tests with Welch's correction. "n.s." refers to differences that are not statistically significant.
(TIF)

**S4 Fig. Quantification of native polyQ40::YFP and polyQ44:YFP protein.** *C. elegans* were exposed for 3d to nanoparticulate matter (nPM), the oxidant paraquat (PQ), or mock exposed to vehicle (M9) starting at the L1 stage followed by 1hr recovery in the case of polyQ40::YFP (strain AM141) (A,B) or at the L4 stage followed by 3d of recovery for polyQ44::YFP animals (strain OG412) (C,D). Total native protein was extracted and native gel electrophoresis was performed in at least biological triplicate. A, C) The ratios of the indicated high molecular weight species to monomers within each lane are shown as averages of biological replicates (n = 5 for polyQ40::YFP and n = 3 for polyQ44::YFP) (bars) and also as individual biological replicates (●, M9; ■, PQ; ▲, nPM). T-tests were performed with Welch's correction. "n.s." refers to differences that are not statistically significant. (B,D) Representative native gels showing in-gel YFP fluorescence with high molecular weight species and monomers indicated.
(TIF)

**S1 Raw images.**
(PDF)

## Acknowledgments

We would like to thank the laboratory of Caleb E. Finch and the University of Southern California for generously providing us with nPM. We would also like to thank members of the Kikis lab, especially Prisha Rajasekaran and Jeremiah Studivant for technical support and also for their careful reading and thoughtful comments and discussion of this manuscript. This research project was supported in part by the Emory University Integrated Cellular Imaging Core and some strains were provided by the *C. elegans Genetics Center* (CGC).

## Author Contributions

**Conceptualization:** Elise A. Kikis.

**Formal analysis:** Elise A. Kikis.

**Funding acquisition:** Elise A. Kikis.

**Investigation:** Bailey A. Garcia Manriquez, Julia A. Papapanagiotou, Claire A. Strysick, Emily H. Green, Elise A. Kikis.

**Methodology:** Elise A. Kikis.

**Project administration:** Elise A. Kikis.

**Supervision:** Elise A. Kikis.

**Visualization:** Elise A. Kikis.

**Writing – original draft:** Elise A. Kikis.

**Writing – review & editing:** Bailey A. Garcia Manriquez, Julia A. Papapanagiotou, Claire A. Strysick.

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
