## [Decision Letter · Decision Letter 0]

1 Dec 2022

PONE-D-22-25226Nanoparticulate air pollution disrupts proteostasis in Caenorhabditis elegansPLOS ONE

Dear Dr. Kikis,

Thank you for submitting your manuscript to PLOS ONE. After careful consideration, we feel that it has merit but does not fully meet PLOS ONE’s publication criteria as it currently stands. Therefore, we invite you to submit a revised version of the manuscript that addresses the points raised during the review process.

We look forward to receiving your revised manuscript.

Kind regards,

Mahesh Narayan, Ph.D.

Academic Editor

PLOS ONE

Journal Requirements:

"We would like to thank the laboratory of Caleb E. Finch and the University of Southern California for generously providing us with nPM. We would also like to thank members of the Kikis lab, especially Prisha Rajasekaran and Jeremiah Studivant for technical support and also for their careful reading and thoughtful comments and discussion of this manuscript. This research project was supported in part by the Emory University Integrated Cellular Imaging Core and some strains were provided by the CGC, which is funded by the NIH Office of Research Infrastructure Programs (P40 OD010440). The content is solely the responsibility of the authors and does not necessarily reflect the official views of the National Institute of Health."

"E.A.K was supported by a Faculty Fellowship from the Appalachian College Association and by two internal grants, a Faculty Development Grant and a Faculty Summer Research Stipend, from the University of the South. B.A.G.M, J.A.P., and C.A.S. were funded by Sewanee Undergraduate Research Fellowships. The CGC, which provides some strains used in this study, and the Emory University Integrated Cellular Imaging Core, which provided microscopy services, were funded by the National Institutes of Health. Funders had no role in study design, data collection and analysis, decision to publish, or preparation of the manuscript."

Additional Editor Comments (if provided):

Dear Dr. Kikis

Please see the reviewers comments and respond accordingly for further consideration of your MS

sincerely

MN

Reviewers' comments:

Reviewer's Responses to Questions

**Comments to the Author**

1. Does the manuscript adhere to the experimental procedures and analyses described in the Registered Report Protocol?

If the manuscript reports any deviations from the planned experimental procedures and analyses, those must be reasonable and adequately justified.

Reviewer #1: Yes

Reviewer #2: Yes

2. If the manuscript reports exploratory analyses or experimental procedures not outlined in the original Registered Report Protocol, are these reasonable, justified and methodologically sound?

A Registered Report may include valid exploratory analyses not previously outlined in the Registered Report Protocol, as long as they are described as such.

Reviewer #1: Yes

Reviewer #2: Yes

3. Are the conclusions supported by the data and do they address the research question presented in the Registered Report Protocol?

The manuscript must describe a technically sound piece of scientific research with data that supports the conclusions. The conclusions must be drawn appropriately based on the research question(s) outlined in the Registered Report Protocol and on the data presented.

Reviewer #1: Yes

Reviewer #2: Yes

4. Have the authors made all data underlying the findings in their manuscript fully available?

Reviewer #1: Yes

Reviewer #2: Yes

5. Is the manuscript presented in an intelligible fashion and written in standard English?

Reviewer #1: Yes

Reviewer #2: Yes

6. Review Comments to the Author

Please use the space provided to explain your answers to the questions above. (Please upload your review as an attachment if it exceeds 20,000 characters)

Reviewer #1: In the manuscript entitled “Nanoparticulate air pollution disrupts proteostasis in Caenorhabditis elegans”, Manriquez et al. evaluate the impact of environmental risk factors such as air pollution on proteostasis. Using well established C. elegans protein conformation disease (PCD) models, the authors examine protein aggregation using visual and biochemical assays and monitor proteotoxicity using behavioral readouts for movement. Based on this analysis, the authors conclude that nanoparticulate matter (nPM) exacerbates protein aggregation and proteotoxicity in the PCD models and has no impact on stress responses in wild type animals. Taken together, the authors suggest that exposure to environmental air pollution may overwhelm the proteostasis network and worsen symptoms associated with Alzheimer’s disease and dementia. Overall, the manuscript is well organized and well communicated. This reviewer concludes that the manuscript is appropriate for publication in PLOS ONE after addressing the issues outlined below.

To support the authors’ conclusions, the following additions to the experimental data are recommended.

1. To better view the visual aggregates that were quantified in Figure 1B, include an inset with higher magnification that depicts the type of aggregate that was scored. Specifically, magnify the area at the end of the arrowheads.

2. It is unclear what the reader should observe in Figure 1C. Addition of arrows to denote what the authors refer to as AB puncta and arrowheads to highlight what the authors refer to as larger aggregates is needed.

3. Supplemental Figure 2 shows the quantification of relative protein levels in each polyQ model. To support the conclusion that protein levels do not change, a representative gel image of the area quantified would be helpful. Specifically, polyglutamine tracts can lead to accumulation of proteins in the stacking gel of SDS PAGE gels. It would be important to note if protein accumulated in the stacking gel and if this was included in the final determination of protein levels. An image of a representative gel/blot to resolve this would be informative.

4. In Supplemental Figure 2, the authors state that there is no statistical difference (text line 241 but do not provide p values. This is relevant for panel C as the relative level between M9 control and nPM look different.

5. Fix line 264. Change AB to polyQ

6. The formatting in Figure 2B and 2C make it difficult to discern the impact of nPM on N2. It appears as though the lines overlap but a statement in the text would help resolve this confusion.

7. In Figure 2B, it is unclear if there is a statistically significant effect of the acute exposure to nPM for strain GMC101. There is an asterisk on the graph but it is unclear what this refers to. Furthermore, this reviewer questions if a t-test is the appropriate statistical analysis for this data set. The authors should clarify since Line 272 does not state if these differences (IT50) were statistically significant.

8. In Figure 3, it is unclear if the high molecular species are normalized to total protein in each lane or to control. Or are the values determined only by comparing intensity of the HMW species vs monomers for each strain. This reviewer recommends calculating relative amounts of HMW in total for each strain.

9. In Supplemental Figure 4, the figure legend states that protein was collected for the Q40 strain immediately instead of after 72 hours, which was the procedure for aggregate counting. Is there a rationale for why the treatment with nPM was not similar to the procedure for aggregate counting?

10. Supplemental Figure 4 shows the quantification of high molecular weight species in the Q40 and Q44 models. A representative native gel image highlighting the area quantified would be beneficial.

The recommendations above ask for additional data and further analysis of the data to strengthen the authors conclusions. This reviewer offers additional suggestions to address within the text to further strengthen the manuscripts conclusion.

1. Have the authors examined the impact of nPM on stress responses in PCD models? The authors demonstrate no impact on N2 animals but the extra challenge of expressing a misfolded disease protein along with the nPM may lead to activation of a stress response.

2. In Supplemental Figure 1A, nPM leads to an unexpected decrease in Q40 aggregates. To explain this surprising result, the authors state that Q35 is a “better sensor” than Q40. The authors should also consider that the treatments are done at different times in the developmental cycle of the worm, which may affect proteostasis demands.

3. In the conclusions section, it would be helpful if the authors offer their insight into why nPM that exacerbates protein misfolding in body wall muscles can suggest deficiencies in protein folding of AB species in neurons leading to dementia.

Reviewer #2: The paper by Garcia Manriquez et al is a well written report that describes a study in C.elegans models to assess the proteotoxicity of nanoparticulates associated with air pollution. The study is technically sound. The methods used appear appropriate and appropriate statistical methods were used to assess the data. One short-coming of the study is uncertainty regarding the potential exposure in vulnerable neurons of human brain relative to the dose of nanoparticulates that the worms were exposed to.

The only technical comment I have is that in Fig 1C, the authors should indicate the number of replicates that were examined to produce the representative images.

7. PLOS authors have the option to publish the peer review history of their article (what does this mean?). If published, this will include your full peer review and any attached files.

Reviewer #1: No

Reviewer #2: No

---

## [Author Response · Author response to Decision Letter 0]

4 Jan 2023

See file titled "response to reviewers."

---

## [Editor Report · Decision Letter 1]

2 Feb 2023

Nanoparticulate air pollution disrupts proteostasis in Caenorhabditis elegans

PONE-D-22-25226R1

Dear Dr. Kikis,

We’re pleased to inform you that your manuscript has been judged scientifically suitable for publication and will be formally accepted for publication once it meets all outstanding technical requirements.

Kind regards,

Mahesh Narayan, Ph.D.

Academic Editor

PLOS ONE
---

## [Editor Report · Acceptance letter]

14 Feb 2023

PONE-D-22-25226R1 

Nanoparticulate air pollution disrupts proteostasis in *Caenorhabditis elegans*

Dear Dr. Kikis:

I'm pleased to inform you that your manuscript has been deemed suitable for publication in PLOS ONE. Congratulations! Your manuscript is now with our production department. 

Kind regards, 

on behalf of

Dr. Mahesh Narayan 

Academic Editor

PLOS ONE